# Shorted Happiness at Work Scale: Psychometric Proprieties of the Portuguese Version in a Sample of Nurses

**DOI:** 10.3390/ijerph20010658

**Published:** 2022-12-30

**Authors:** Sofia Feitor, Teresa Martins, Elisabete Borges

**Affiliations:** 1Nursing School of Porto, ESEP, 4200-072 Porto, Portugal; 2Center for Health Technology and Services Research—CINTESIS@RISE, 4200-450 Porto, Portugal

**Keywords:** happiness at work, scale, psychometrics, nurses, occupational health

## Abstract

In the last decades, happiness at work has been studied due to social changes; increased workload; stress; and, more recently, the COVID-19 pandemic. Happiness at work is considered an umbrella concept as it covers individual and organizational aspects of working life. The aim of this study is to analyze the psychometric properties of the Shorted Happiness at Work Scale (SHAW) in a sample of Portuguese nurses. A cross-sectional study with 113 Portuguese nurses, from one of the islands of the Azores, was selected through a convenience sample. A sociodemographic/professional questionnaire and the SHAW scale were applied. Through the CFA, the SHAW trifactorial model was tested according to its theoretical reference, having obtained a general tolerable adjustment index. After respecification of the model through the correlation of the errors of two items, a better adjustment was obtained, but the RMSEA value remains problematic. Additionally, the values of the coefficient of internal consistency were indicative of good fidelity. The analysis of the psychometric characteristics of the SHAW scale, in the sample of Portuguese nurses, suggests a theoretical adaptation to the model of happiness at work.

## 1. Introduction

In recent decades, work changes and new evidence have prompted organizations and managers towards new theories and practices. Happiness at work has emerged as a new subject in the literature, in association with positive psychology [1,2]. More recently, the COVID-19 pandemic has brought new challenges, forcing companies, employers, and workers to adapt to economic, social, and health difficulties. In fact, in times of crisis such as the one we are currently experiencing, the flexibility of companies is crucial, in order to generate learning and growth [3,4].

Happiness at work is defined as a set of individual factors, encompassing stable aspects (such as emotions and moods) and transitory aspects (personal dispositions and attitudes), and organizational factors [5]. To include all of these factors, Fisher proposed a measurement model through a combination of concepts, namely, engagement (involvement at work), job satisfaction (judgment about the job’s characteristics), and affective organizational commitment (emotional attachment and identification with the company). Therefore, two attitudinal constructs (job satisfaction and affective organizational commitment) and a motivational construct (engagement) are combined, providing a more stable image over time [5].

Other authors [6,7,8] define happiness at work by focusing on the existence of a greater number of positive work experiences to the detriment of negative ones, in the experience of meaning at work and in the maximization of professional performance.

Although it is a relatively recent concept, there is evidence of the positive consequences of investing in happiness at work in organizations [4]. At the individual level, it is possible to highlight the increase in energy levels, happiness in personal life, self-confidence, professional performance, and health [9]. At the organizational level, productivity increases and employee turnover, as well as overall company costs, decrease [9].

Furthermore, the evaluation and enhancement of happiness at work are important tools for managers and organizations, facilitating the attraction of creative people [10], being the competitive key to securing the best workers [4,11].

Accordingly, the need to approach the concept of happiness at work in a multidisciplinary way is necessary, increasing the number of studies in the area [12] based on reliable, complete, and objective scales [10].

The measurement of the concept of happiness at work will always depend on the definition adopted by the author(s). However, researchers try to provide an operational definition of the construct in order to measure it. Some scales for measuring happiness at work have appeared in the literature, such as the happiness at work scale [8], the questionnaire for the evaluation of happiness at work [13], and the job design happiness scale [14]. There is also the happiness at work scale (HAW) [15], which has a shortened version, the shorted happiness at work scale (SHAW) [10].

The HAW scale [15] was built based on the Fisher’s definition of happiness at work [5]. Therefore, researchers can capture work passion and enthusiasm, workers’ evaluations of working conditions, and feelings of belonging towards the company. That said, the authors combined three widely known and validated scales to create the HAW scale. Regarding engagement, the scale used was the Utrecht work-enthusiasm scale [16], which has three subscales: vigor, dedication, and absorption. The job satisfaction index was the scale chosen to globally assess job satisfaction [17]. Finally, affective organizational commitment was measured using the Allen and Meyer scale, which encompasses affective, normative, and continuity commitment [18]. Yet, the scale remained with 31 items [15].

After the process of the scale’s reduction, the SHAW scale has been left with nine items, with each dimension (engagement, job satisfaction, and affective organizational commitment) having three corresponding items. The scale is Likert-type, scored from 1 (strongly disagree) to 7 (strongly agree), and the higher the score, the higher the level of happiness at work [10].

Nursing is the largest health care occupation in the world. The enhancement of nurse’s happiness at work will mean better quality care and a general improvement in the population’s health [4,19].

Due to the COVID-19 pandemic, health professionals faced difficult challenges, such as intense workloads, the risk of infection and death, inadequate resources (mainly individual protection), and social discrimination, in addition to having to take care of their family and friends [20]. This led to a significant decrease in their mental health, with increased levels of burnout, absenteeism, and quality and safety of care, highlighting the importance of occupational health services [19,21].

Thus, the measurement and evaluation of happiness at work of these professionals becomes very important, to delineate effective and personalized strategies for each reality [4], so that retention of workforce in health institutions is enhanced.

Accordingly, the aim of this study is to analyze the psychometric properties of the SHAW scale in a sample of Portuguese nurses.

## 2. Materials and Methods

### 2.1. Study Design and Population

A cross-sectional study was carried out with Portuguese nurses, working in a hospital unit and in a primary health care unit, on an island in the Azores archipelago, for at least six months. Nurses were invited to participate in person by the researcher, who provided information about the study. From a total of 167 nurses, a convenience sample of 113 was obtained, corresponding to a total adherence of 67.7%.

### 2.2. Measures

A questionnaire for sociodemographic and professional characterization of the sample (including gender, age, marital status, children, academic qualifications, length of professional experience, professional category, working hours, and type of relationship with the institution) and the SHAW scale [10] were applied, in September 2020.

The SHAW scale includes nine items, rated on a 7-point Likert scale, ranging from 1 (strongly disagree) to 7 (strongly agree), organized into three dimensions: engagement (items 1, 2, and 3), job satisfaction (items 4, 5, and 6), and affective organizational commitment (items 7, 8, and 9). Engagement is characterized by enthusiasm and passion at work and positive mental states (vigor, dedication, and absorption), job satisfaction is related to a worker’s judgments about working conditions, and affective organizational commitment is related to the relational and emotional part of the work and the compromise that arises between the worker and the organization. The higher the score, the happier the individual is at work [10]. The SHAW scale has obtained good psychometric properties in several studies [10,22,23].

### 2.3. Statistical Analysis

For data analysis, the SPSS^®^ version 28.0 program (IBM, Armonk, NY, USA) was used. Descriptive statistics were used, specifically absolute and relative frequencies, and mean and standard deviation. The internal consistency of the scale and dimensions were performed using Cronbach’s Alpha. A significance level of 0.05 was considered.

Confirmatory factorial analyses (CFA) were used through the software program IBM-AMOS version 28, to test the three-factor model of SHAW.

In order to analyze the significance of all items of each factor, to scale the factors, the variance of the four factors was set to 1. The covariance matrix was considered as the input, adopting the maximum likelihood method estimation. The existence of outliers was evaluated. The normality was evaluated by the coefficient of skewness and univariate and multivariate kurtosis.

The quality of the model fit was performed according to the indices and respective reference values. Thus, a lower RMR value suggests a better fit, with RMR = 0 indicating a perfect fit. Regarding the CFI indices, values greater than 0.90 are indicative of a good fit as, for the RMSEA values, values up to 0.10 are acceptable [24]. The local adjustment was evaluated by the Mahalonobis squared distance, the factor weights, and the individual reliability of the items. The comparative fit index (CFI), the root mean squared residual (RMR), the root mean square error approximation (RMSEA), their confidence intervals (CI), and the modification indices were considered. The model adjustment considered the theoretical considerations.

### 2.4. Procedures

The present study was approved by the Directing Board and by the Ethics Committee of the Hospital Unit and Health Unit (Distribution Reports No. 1018 and 1891). After all of the approvals were given, contact was established by the researcher with the nurses in charge of every department, presenting the purpose of the study and the data collection procedures. The data collection instrument, on paper, was delivered to the nurses in charge of each department, as previously established, as well as envelopes. After completing the questionnaire, the participant placed it in a closed envelope. A period of 15 days was established for the instrument’s collection, which was carried out by the researcher and without direct contact with the participants. All of the participants also gave their consent by signing the informed consent declaration.

## 3. Results

The sample of 113 nurses was mostly female (89.4%), aged 41 years or older (46.9%), with a partner (75.2%), and with children (68.1%). They had a degree (73.5%), an average professional experience of 18 years (SD = 10), a permanent employment contract (97.3%), a fixed working schedule (63.7%), and were generalist nurses (72.6%). These and other sociodemographic and professional variables are presented in Table 1.

Table 2 shows the mean, standard deviation, and internal consistency values (Cronbach’s Alpha) and the SHAW scale dimension items.

The Portuguese nurses showed moderate levels of happiness at work (M = 4.25; SD = 1.05).

The values of the Cronbach’s Alpha coefficient for the total scale and its dimensions are representative of good internal consistency and are like those obtained by other authors [10,22,23], although with relatively lower values.

We tested whether the three-factor SHAW model fitted the theoretical model advocated by Salas–Vallina and colleagues [10].

The results showed a tolerable fit of the measurement model, at most fit indices, with values of X2 (24, *n* = 113) = 88.757, *p* < 0.0001, RMR = 0.256, and CFI = 0.893, but with a poor RMSEA = 0.155, 90% CI [0.121, 0.190], and *p* (RMSEA ≤ 0.05) < 0.0001.

An analysis of the parameter estimates revealed that the factors were highly correlated and that all standardized indices saturate the respective factor, with factor weights ranging between 0.32 and 0.92 (all *p* < 0.001).

The standardized residuals were analyzed to identify possible sources of problems that could justify the bad fit and the suggested modification rates. Through these, a respecification of the model was carried out with the correlation between the errors related to the items J_S2 and J_S3, after debating the content of the items and there being a consensus of the authors about their interaction. In fact, in our Portuguese context, promotion in the nursing professional career is, as a rule, associated with a salary increase, hence the covariance between the two observed variables.

Accordingly, the new model started to obtain adjustment indices compatible with a reasonable adjustment (Figure 1), with the following results: X2 (23, *n* = 113) = 59.904, *p* < 0.0001, RMR = 0.214, CFI = 0.939, RSMEA = 0.120, 90% CI (0.083, 0.157), and *p* (RMSEA ≤ 0.05) = 0.002.

By analyzing the matrix of standardized correlations between the latent variables, it was found that the three dimensions are correlated, with a strong correlation between engagement and job satisfaction being visible.

## 4. Discussion

The three-factor SHAW model showed satisfactory adjustment indices, after re-specification of the model with the correlation of errors in items 2 and 3 of job satisfaction. Only the RMSEA value, essentially its upper limit, exceeds the recommended value. However, it is known that this index is influenced by the sample size and by model degrees of freedom [25]. This index tends to obtain critical values in models with few degrees of freedom, so we must take these aspects into consideration and evaluate its behavior in future studies.

Therefore, the structure of the Portuguese version of SHAW scale overlaps the measurement model, giving consistency to the results found. The values of the internal consistency of the dimensions and of the total scale suggest that the measure presents good fidelity. These psychometric characteristics, associated with the fact that the scale is brief, mean that the measure has advantages for its use in a research context. As stated by the authors of the original version of the scale, this is an accessible and fast instrument, in a world where positive management is growing exponentially [10]. Furthermore, they claim that it has great statistical potential to assess positive attitudes at work, which leads to new forms and possibilities of research [10].

Other studies with the SHAW scale suggest results in agreement with those of the present study. In India, Rastogi [26] studied happiness at work in Eastern countries, having previously validated the SHAW scale, in a sample of 226 workers in the public and private sector. The author obtained psychometric properties similar to those obtained by the authors of the scale [10], having obtained a CFI value slightly higher than that of the present study (0.989). The author found moderate correlations between the dimensions of the scale, with the strongest relationship between the dimension of engagement and job satisfaction (0.60) [26], as observed in the present study.

Atan et al. [1] applied the SHAW scale to 271 female workers in four- and five-star hotels in northern Cyprus, and they also obtained good psychometric properties. Their exploratory-factor analysis corroborates the existence of three dimensions, and the factors explained 85.012% of the total variance. They obtained a CFI of 0.923, a value very close to that obtained in the present study. As for the factor loadings of the items, these are higher than 0.40, and all of their correlations are significant.

Bilginoğlu and Yozgat [27] also tested the factor validity of the SHAW scale, in a sample of 276 Turkish workers, using exploratory factor analysis, internal consistency, and test–retest reliability analysis. Its exploratory factor analysis explained 84.9% of the variance. Cronbach’s Alpha coefficient for the global scale was 0.89, and the test–retest reliability coefficient was 0.72. The authors suggest that the SHAW scale is reliable and valid for the Turkish population [27].

Thus, the assessment of happiness at work can be an important diagnostic tool, with a view to improving the management of organizations, directing the training of managers to a more human perspective [15]. Since it is an emerging field of investigation, it is important that it is studied [28].

In addition to being a new concept, the assessment of happiness at work does not have many valid measuring instruments. Gabini [29] reflects on why this happens. First, it is because other better-known constructs are preferred to the construct of happiness at work, such as quality of work life and job satisfaction. Other researchers choose to somehow modify pre-existing scales of global happiness, while others combine some scales in an attempt to combine as many concepts as possible, which, in their opinion, are covered by happiness at work [29].

The SHAW scale in the present study proved to be reliable and acceptable, and it seems to be a valid instrument regarding the assessment of happiness at work, photographing it in a comprehensive way through its dimensions and seeking to provide tools to improve quality of life at work, as also mentioned by the authors of the scale [10].

Creating positive work environments will lead to fostering interpersonal connections and increasing the effectiveness of each worker/organization. Measuring happiness at work can bring this much-needed human perspective to today’s institutions, raising levels of professionalism [15]. The focus should be, firstly, on the human capital of companies as this is the mechanism that drives growth and innovation [10,30].

Assessing the levels of happiness at work of nurses and understanding the factors that influence them is central to the design of strategies that seek to increase this same happiness, retain professionals, increase productivity, and provide better care for the general population. This will bring innovation and relevant scientific contributions to the nursing profession and discipline and to the entire community [4].

Some of the limitations of the study were related to the sample size. The fact that the model has a correlation of two errors must be considered, and it would be desirable, in future investigations, to verify if this characteristic is maintained.

The fact is that the construct of happiness at work has still been little studied, and the SHAW scale is quite recent and has few validations for other populations, so it is more complicated to use it compare data and psychometric properties.

Further research should analyze the psychometric properties of the SHAW scale in other occupational samples and in other countries. This will increase the knowledge regarding happiness at work and make the SHAW scale a wider-known instrument in the academic field. Therefore, happiness at work itself will improve in other work environments since its evaluation is the first step to action.

## 5. Conclusions

The analysis of the psychometric characteristics of the SHAW scale in the sample of Portuguese nurses, after adjustment, suggests a theoretical adequacy to Fisher’s (2010) model of happiness at work, which encompasses the dimensions of engagement, job satisfaction, and affective organizational commitment. For that reason, it is a tool with the necessary metric qualities to be applied in the Portuguese context.

## Figures and Tables

**Figure 1 ijerph-20-00658-f001:**
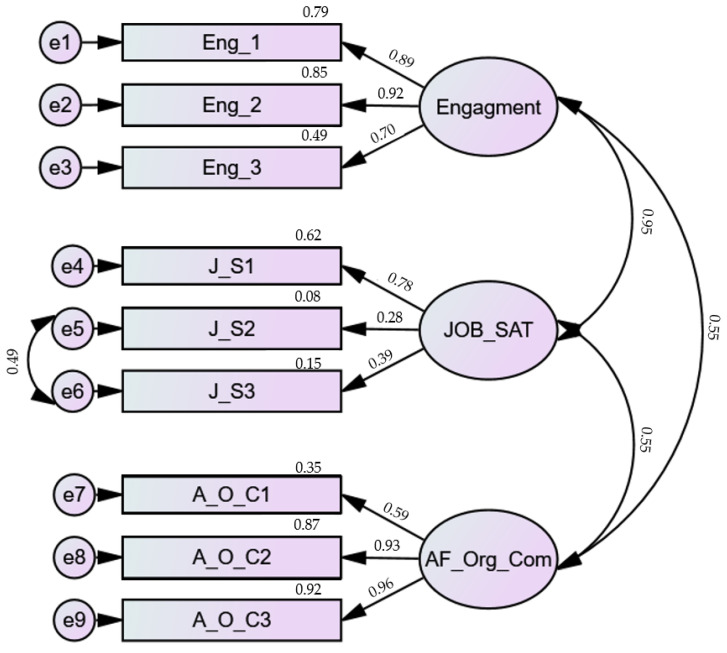
Final model of shorted happiness at work scale with a sample of Portuguese nurses.

**Table 1 ijerph-20-00658-t001:** Participants characteristics.

Participants Characteristics	*n* (%)
Sex	
Male	11 (9.7)
Female	101 (89.4)
No answer	1 (0.9)
Age	
≤40 years	54 (47.8)
≥41 years	53 (46.9)
No answer	6 (5.3)
Marital status	
With partner	85 (75.2)
Without partner	28 (24.8)
Children	
Yes	77 (68.1)
No	36 (31.9)
Academic habilitations	
Degree	83 (73.5)
Postgraduate/Master’s degree/Doctoral degree	29 (25.7)
No answer	1 (0.9)
Professional Experience (years)	
<16 years	49 (43.4)
≥16 years	64 (56.6)
Professional Category	
Generalist nurse	82 (72.6)
Specialist nurse	20 (17.7)
Nurse manager	10 (8.8)
No answer	1 (0.9)
Work shift	
Fixed shifts	72 (63.7)
Rotating shifts	40 (35.4)
No answer	1 (0.9)
Employment contract	
Precarious	2 (1.8)
Permanent	110 (97.3)
No answer	1 (0.9)

**Table 2 ijerph-20-00658-t002:** McDonald’s, Cronbach’s Alpha, mean, standard deviation, SHAW’s items, and dimensions.

Scale/Dimensions (1–7)	Items	McDonald’s Omega Coefficient		Cronbach’s Alpha α	M	DP
		ω	95% IC			
Engagement	At my job, I feel strong and vigorous	0.875	0.835–0.915	0.866	4.77	1.30
	I am enthusiastic about my job					
	I get carried away when I am working					
Work Satisfaction	How satisfied are you with the nature of the work you perform?	0.683	0.579–0.786	0.631	3.27	1.10
	How satisfied are you with the pay you receive for your job?					
	How satisfied are you with the opportunities that exist in this organization for advancement (promotion)?					
Affective Organizational Commitment	I would be very happy to spend the rest of my career with this organization	0.866	0.822–0.910	0.851	4.69	1.40
	I feel emotionally attached to this organization					
	I feel a strong sense of belonging to my organization					
SHAW total		0.867	0.830–0.904	0.867	4.25	1.05

## Data Availability

The data are not publicly available due to privacy issues. Requests to the datasets should be directed to Sofia Feitor, sophiefeitor@gmail.com.

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
