# Peer review of "Shorted Happiness at Work Scale: Psychometric Proprieties of the Portuguese Version in a Sample of Nurses"

_ijerph, 2022, doi:10.3390/ijerph20010658_

Round 1

Reviewer 1 Report

Dear authors, thank you for the opportunity to read your interesting research.

The introduction describes the relevance and scientific knowledge of the issue regarding the tools for assessing happiness at work, and also defines the purpose of this study.

Materials and methods

1. It is proposed to structure this section: Procedure, Sample, Tool, Statistical data analysis. The content of this section presented by the authors is proposed to be redistributed among these subsections.

results

2. I would like to see in the table the distribution of surveyed employees regarding socio-demographic characteristics (not just a description of the prevailing groups).

3. In the discussion of the results, could the authors supplement the analysis of the results of the sample of nurses (analysis of the severity of their happiness at work), this could also be of interest to readers.

4. State clearly in the discussion of the results the limitation of the study and further research directions.

The conclusion is correct and reflects the main result of the work.

With respect for your work and best wishes, reviewer

Author Response

Dear Reviewer 1,

We would like to thank you for your observations and suggestions.

The comments are all valuable and very helpful for revising and improving our paper, as well as of important guiding significance to our research.

In this context, changes were made throughout the article, highlighted with the “track changes” feature of Microsoft Word and described in the end of the document.

We wish you a very merry Christmas.

Kind regards,

Sofia Feitor.

Reviewer 2 Report

I consider that the topic discussed in the article is of interest as all those studies that help improve the work environment and the satisfaction of workers in organizations. Therefore, I understand that the subject addressed is topical and important.

Regarding the title, I think that the sample in which the psychometric characteristics of the scale is studied should be specified.

The justification of the study well elaborated, explaining the topics and defining the variables under study.

As for the method, the population and the sample used are clearly described, as well as the instruments used to obtain the data. However, the procedure should be further developed to make it much clearer. This section could also be organized, to gain clarity for the reader, following the classic epigraphs of an investigation, sample, instruments and procedure.

The results are clear and the techniques for obtaining these are adequate.  Similarly, the discussion is well justified and based on the results obtained.

Finally, I agree with the limitations outlined by the authors, although I believe that there are many more limits than those indicated by them. I think they should review their research and think about all the limitations that their study has.

It seems to me that the article meets the minimum requirements to be published with the proposed modifications or even without them.

Author Response

Dear Reviewer 2,

We would like to thank you for your observations and suggestions.

The comments are all valuable and very helpful for revising and improving our paper, as well as of important guiding significance to our research.

In this context, changes were made throughout the article, highlighted with the “track changes” feature of Microsoft Word and described in the end of the document.

We wish you a very merry Christmas.

Kind regards,

Sofia Feitor.

Reviewer 3 Report

Thanks for submitting the manuscript. There are several major issues related to the research methodology and data analysis methods:

1) The CFA model has been adjusted with correlating the error terms. The existing literature is strongly discourage it (Hermida, 2015). Please provide the justifications.

2) The IBM AMOS is using Maximum Likelihood for the CFA estimator, which was criticized in the existing literature (Li, 2016). The authors should use other estimators, such as DWLS.

3) Please clarify the cut-off values for each model fit indices in the methods section.

4) In addition to the Cronbach’s alpha, please also report the McDonald’s Omega coefficient and hierarchical value (Beland et al., 2017; Gignac et al., 2019).

5) After correlating the error terms, the reported CFA results “X2 (23, N = 113) = 59,904, p < 0,0001, RMR = 0,214, CFI = 0,939 , RSMEA = 0,120, 159 90% CI [0,083, 0,157], p (rmsea≤ 0,05) = 0,002” still failed to fulfil the minimum requirement for adequate model fit, as RMR and RMSEA are over 0.08, RNR (Kline, 2005). In other words, the data did not support the adapted scale with good construct validity.

6) The evaluation of concurrent validity is missing.

Hence, in view of the about issues, I do not think the current form of manuscript is suitable for publication.

References

Beland, S., Cousineau, D., & Loye, N. (2017). Using the Mcdonald's Omega Coefficient instead of Cronbach's Alpha [Article]. Mcgill Journal of Education, 52(3), 791-804. <Go to ISI>://WOS:000437472500013

Gignac, G. E., Reynolds, M. R., & Kovacs, K. (2019). Digit Span Subscale Scores May Be Insufficiently Reliable for Clinical Interpretation: Distinguishing Between Stratified Coefficient Alpha and Omega Hierarchical [Article]. Assessment, 26(8), 1554-1563. https://doi.org/10.1177/1073191117748396

Hermida, R. (2015). The problem of allowing correlated errors in structural equation modeling: concerns and considerations. Computational Methods in Social Sciences (CMSS), 3(1), 05-17. https://EconPapers.repec.org/RePEc:ntu:ntcmss:vol3-iss1-15-005

Kline, R. B. (2005). Principles and practice of structural equation modeling (2 ed.). New York: Guilford Press.

Li, C. H. (2016). Confirmatory factor analysis with ordinal data: Comparing robust maximum likelihood and diagonally weighted least squares [Article]. Behavior Research Methods, 48(3), 936-949. https://doi.org/10.3758/s13428-015-0619-7

Author Response

Dear Reviewer 3,

We would like to thank you for your observations and suggestions.

The comments are all valuable and very helpful for revising and improving our paper, as well as of important guiding significance to our research.

In this context, changes were made throughout the article, highlighted with the “track changes” feature of Microsoft Word and described in the end of the document.

We wish you a very merry Christmas.

Kind regards,

Sofia Feitor.
